# ASMa: Asymmetric Spatio-temporal Masking for Skeleton Action Representation Learning

**Aman Anand**  *aman.anand@queensu.ca*
*School of Computing*
*Queen's University*

**Amir Eskandari**  *amir.eskandari@queensu.ca*
*School of Computing*
*Queen's University*

**Elyas Rahsno**  *elyas.rashno@queensu.ca*
*School of Computing*
*Queen's University*

**Farhana Zulkernine**  *farhana.zulkernine@queensu.ca*
*School of Computing*
*Queen's University*

**Reviewed on OpenReview:** *https://openreview.net/forum?id=kIFo1q3VMS*

## Abstract

Self-supervised learning (SSL) has shown remarkable success in skeleton-based action recognition by leveraging data augmentations to learn meaningful representations. However, existing SSL methods rely on data augmentations that predominantly focus on masking high-motion frames and high-degree joints such as joints with degree 3 or 4. This results in biased and incomplete feature representations that struggle to generalize across varied motion patterns. To address this, we propose Asymmetric Spatio-temporal Masking (ASMa) for Skeleton Action Representation Learning, a novel combination of masking to learn a full spectrum of spatio-temporal dynamics inherent in human actions. ASMa employs two complementary masking strategies: one that selectively masks high-degree joints and low-motion, and another that masks low-degree joints and high-motion frames. These masking strategies ensure a more balanced and comprehensive skeleton representation learning. Furthermore, we introduce a learnable feature alignment module to effectively align the representations learned from both masked views. To facilitate deployment in resource-constrained settings and on low-resource devices, we compress the learned and aligned representation into a lightweight model using knowledge distillation. Extensive experiments on NTU RGB+D 60, NTU RGB+D 120, and PKU-MMD datasets demonstrate that our approach outperforms existing SSL methods with an average improvement of 2.7–4.4% in fine-tuning and up to 5.9% in transfer learning to noisy datasets and achieves competitive performance compared to fully supervised baselines. Our distilled model achieves 91.4% parameter reduction and 3× faster inference on edge devices while maintaining competitive accuracy, enabling practical deployment in resource-constrained scenarios.

## 1 Introduction

Skeleton-based human action recognition has gained significant attention in human-machine interaction in various areas such as healthcare (Ren et al., 2024), security (Liu et al., 2024; Ruiz-Santaquiteria et al., 2024),

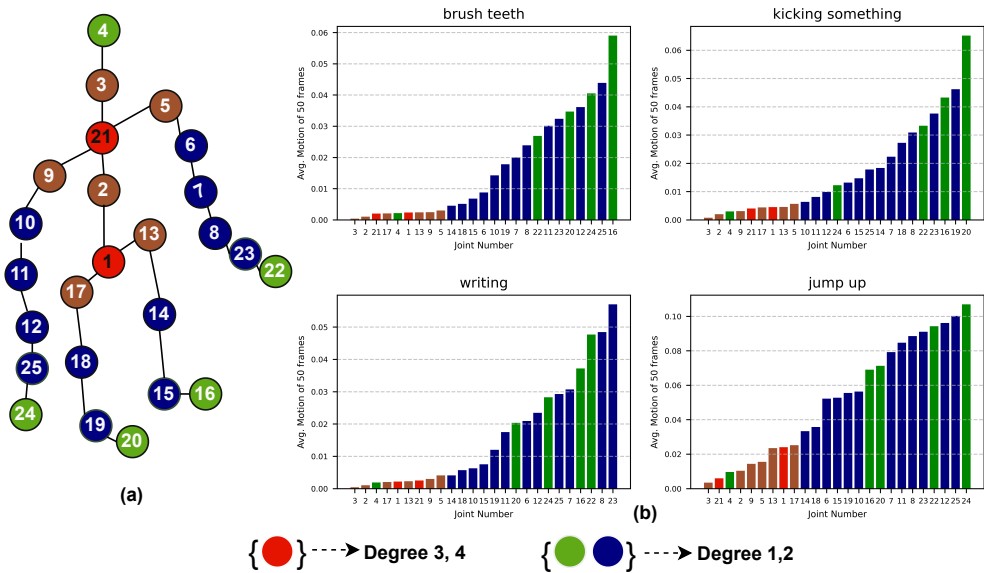

Figure 1: (a) Skeleton graph with joints color-coded by degree centrality. (b) Bar plots show the average motion intensity of each joint for different actions across 50 frames. Through this analysis, we observe that low-degree joints and their adjacent joints exhibit high motion across actions.

and sports analysis (Qi et al., 2018). A skeleton is a simple representation of the human body using a graph structure, where nodes represent the body joints and edges represent the bones. In recent years, many graph-based supervised learning approaches have been introduced for action recognition such as the spatio-temporal graph convolution network (GCN) (Yan et al., 2018), and Dynamic GCN (Wang et al., 2019). Although fully supervised frameworks have shown great promise, they rely on human-annotated datasets. To address this, researchers have proposed self-supervised learning (SSL) frameworks (Zbontar et al., 2021; Grill et al., 2020; Oquab et al., 2023; He et al., 2022; Chen et al., 2020a; He et al., 2020) to learn meaningful representations from the data and achieved comparable performance to supervised frameworks. Compared to conventional RGB or video-based representations, skeleton-based learning offers several advantages. Skeleton data are computationally efficient due to their low-dimensional joint coordinates, robust to variations in lighting, viewpoint, or background, and explicitly encode motion dynamics through inter-joint relationships. These properties make skeleton representations particularly suitable for efficient and interpretable action understanding in real-world settings.

Traditional SSL frameworks focus on RGB images and formulate the pretext (Zbontar et al., 2021; Grill et al., 2020; Oquab et al., 2023) task as learning similar representations for differently augmented views of the same image or predicting masked patches (He et al., 2022) of the image. By adopting these data augmentations, recent studies introduced innovative approaches (Zhou et al., 2023; Lin et al., 2020; Yan et al., 2023) to capture spatio-temporal relationships in skeleton data. In general, skeleton-based SSL methods leverage data augmentation such as masking sets of body joints to learn spatial information and random video frames to understand temporal relationships. While effective, these augmentations often emphasize limited body regions, potentially overlooking the full spectrum of spatio-temporal dependencies necessary for generalizable action recognition. The selective masking can lead to biased representation learning which reduces the generalizability of the model in varying motion patterns across different actions.

Through a simple analysis as illustrated in Figure 1, we observed the contrasting role of joint degree (number of connected nodes) and motion intensity (average motion of 50 frames measured from joint displacement) across different actions. The *High-degree* joints, such as the spine (joints 1 and 21), along with adjacent joints, act as structural stabilizers and exhibit minimal motion. In contrast, the low-degree joints, along with some adjacent joints, show significantly higher motion. This observation suggests that a balanced approach to skeleton representation learning should incorporate both types of motion dynamics to enhance model generalization.

Based on this observation, we aim to learn effective skeleton representations by designing a masking strategy that balances the dynamics of both high-degree joints with low motion and low-degree joints with high motion. We propose an Asymmetric Spatio-temporal Masking (ASMa) for skeleton action representation learning, where we train the two ST-GCN-based encoders with distinct masking strategies. In addition, we introduce a feature alignment module to effectively combine the diverse representations learned by both encoders for downstream tasks. Inspired by the success of the Barlow Twins SSL framework (Lin et al., 2020), we design our model to be a triplet-stream architecture comprising an anchor, a spatial, and a temporal stream that shares a common ST-GCN backbone as an encoder and a Barlow Twins head to learn a cross-correlation matrix between the anchor and the other streams. All three streams are learned simultaneously. While the two encoder architectures lead to richer and more diverse representations, it comes with an inevitable cost of higher inference-time, latency, and memory usage, which require more resources and limit deployability on edge devices. To mitigate this, we propose a lightweight model that learns from the combined representation via knowledge distillation (KD). We then evaluate both encoders with the feature alignment module on NTU60 (Shahroudy et al., 2016), NTU120 (Liu et al., 2019), and PKU-MMD (Liu et al., 2020a) datasets for action classification as a downstream task. To summarize, our main contributions are as follows.

1. We propose ASMa, a novel masking technique that enhances skeleton-based self-supervised representation learning by applying two complementary masking strategies: one focusing on high-degree joints with low motion and the other focusing on low-degree joints with high motion.

2. We introduce a feature alignment module that effectively integrates the representations learned from the above two masking strategies. Through extensive experiments on NTU RGB+D 60, NTU RGB+D 120, and PKU-MMD, we demonstrate consistent performance improvements over prior SSL methods, achieving 1–3% gains in linear probing, 2.7–4.4% in fine-tuning, and 4–6% in transfer learning settings.

3. We also propose a lightweight model using knowledge distillation (KD) that significantly reduces computational complexity while maintaining competitive performance. Additionally, we uncover a novel insight where a student distilled from a linear-probed teacher can surpass the performance of the teacher itself, which demonstrates that generalizable representations can leverage self-supervised distillation.

The rest of the paper is organized as follows: Section 2 reviews related work, Section 3 presents our ASMa methodology including asymmetric masking and feature alignment, Section 4 describes datasets and implementation details, Section 5 provides experimental validation and ablation studies, and Section 6 concludes with future directions.

## 2 Related Work

### 2.1 Skeleton-Based Action Recognition

Early methods for skeleton-based action recognition relied on handcrafted features or statistical models like HMMs (Xia et al., 2012) and Lie group representations (Vemulapalli et al., 2014), but lacked generalization. Deep learning approaches using RNNs, CNNs, and LSTMs improved temporal modeling (Ke et al., 2017), though struggled with long-range dependencies. Graph convolutional networks (GCNs) offered a structural advantage, with ST-GCN (Yan et al., 2018) modeling spatio-temporal relationships over skeleton graphs. Follow-up works improved this with adaptive graphs (Chen et al., 2021), motion shifts (Cheng et al., 2020), and disentangled convolutions (Duan et al., 2022), primarily in supervised settings.

### 2.2 Self-Supervised Representation Learning

Self-supervised learning (SSL) has shown promises in visual domains by learning representations from unlabeled data. Early pretext-task-based methods predicted rotations (Gidaris et al., 2018), spatial context (Noroozi et al., 2017), or masked patches (Pathak et al., 2016). Contrastive methods like MoCo (He et al., 2020) and SimCLR (Chen et al., 2020b) improved feature quality by comparing positive and negative pairs, while BYOL (Grill et al., 2020) and SimSiam (Chen & He, 2021) removed the need for negatives. Barlow Twins (Zbontar et al., 2021) further simplified training by encouraging cross-correlation between views.

Masked modeling techniques like MAE (He et al., 2022) shifted focus to reconstruction-based SSL, effective for modeling spatial and temporal structures.

## 2.3 Self-Supervised Skeleton Representation Learning

Recent SSL approaches for skeleton-based data adapt visual SSL ideas to motion signals. Contrastive frameworks like CrosSCLR (Li et al., 2021) and AimCLR (Guo et al., 2022) leverage inter-view and motion-invariant learning, but often require large memory banks or batch sizes. Negative-sample-free approaches such as (Moliner et al., 2022) and PSTL (Zhou et al., 2023) extend Barlow Twins to skeletons, using triplet streams and partial masking. However, many of these methods bias learning toward high-motion or high-degree joints, potentially neglecting complementary body dynamics. Masked modeling has recently gained attention in this domain. SkeletonMAE (Yan et al., 2023) reconstructs missing joints, but focuses mainly on spatial masking and does not explicitly model motion dynamics. Our work differs by proposing an asymmetric spatio-temporal masking strategy guided by joint degree and motion statistics. Unlike prior works, ASMa encourages learning from both dynamic and stable joint patterns, promoting balanced representation learning.

# 3 Methodology

## 3.1 Overview

Figure 2 presents an overview of our proposed ASMa framework, which consists of three stages: **(i)** Pretraining, **(ii)** Downstream Evaluation, and **(iii)** Knowledge Distillation. In the pretraining stage, we employ two ST-GCN-based encoders, $f_\theta$ and $f_\phi$, each trained using distinct asymmetric spatio-temporal masking strategies. Each encoder processes three streams, namely **anchor**, **spatial**, and **temporal**, simultaneously after applying augmentations and masking as shown in Figure 2(b) and 2(c). During downstream evaluation, a feature alignment module fuses the representations learned by $f_\theta$ and $f_\phi$ into a unified embedding for classification. To support efficient deployment, a lightweight student encoder $f_s$ is trained to distill knowledge from the frozen ASMa teacher.

## 3.2 Masking Strategy

We design an asymmetric masking strategy that operates along both spatial and temporal views of skeleton sequences. All masking strategies are illustrated in Figure 2(a) and defined as follows:

**Notation.** A skeleton sequence is represented as $x \in \mathbb{R}^{C \times T \times V}$, where $C$ is the number of channels (typically 3 for 3D coordinates), $T$ is the number of temporal frames, and $V$ is the number of joints. We apply random augmentations $\mathcal{T}$ (e.g., crop, rotation, flip) to produce views $x'$ and $\hat{x}$, which are then masked to produce spatial and temporal views.

### 3.2.1 Spatial Masking (Joint Selection)

**High-Degree Spatial Masking (HDSM).** To prioritize the skeleton joints with high degrees joints (e.g. spine) in the skeleton to be masked, we define the masking probability $p_v$ for each joint $v$ proportional to its degree centrality $d_v$ :

$$p_v^{(H)} = \frac{d_v}{\sum_{u=1}^{V} d_u}, \tag{1}$$

where $d_v$ is the degree of joint $v$ and the superscript $(H)$ denotes high-degree joint masking. Joints are sampled for masking according to $p_v^{(H)}$ to produce the joint masked view $x_j^\phi$.

**Low-Degree Spatial Masking (LDSM).** To capture complementary dynamics from peripheral joints (e.g., hands, feet), we define:

$$p_v^{(L)} = 1 - p_v^{(H)}, \tag{2}$$

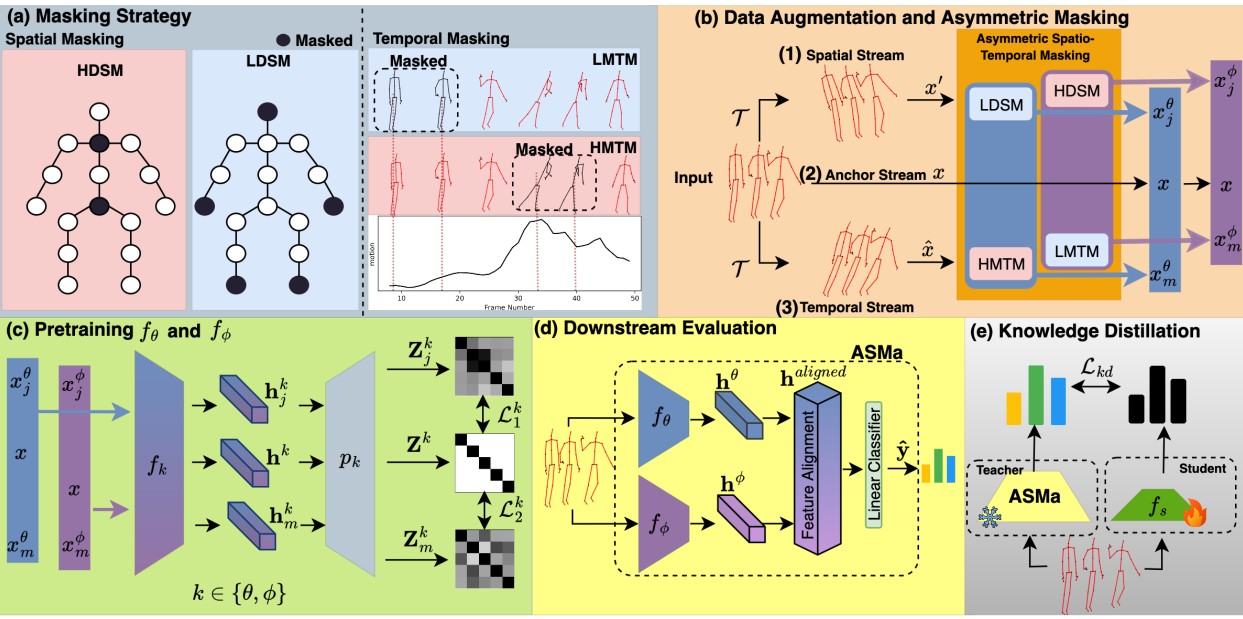

Figure 2: Overview of the ASMa framework. (a) Shows High-Degree Spatial masking (HDSM) and Low-Degree Spatial masking (LDSM) for joint and High-Motion Temporal masking (HMTM) and Low-Motion Temporal masking (LMTM) for frames. (b) Inputs are split into 3 streams with random augmentation $\mathcal{T}$ and masked asymmetrically along spatial and temporal stream. (c) Each encoder processes the triplet streams and is trained using Barlow Twins loss. (d) Learned features from both encoders are fused via a feature alignment module for downstream evaluation. (e) ASMa-Distill: A lightweight student model learns from the frozen ASMa teacher logits.

where the superscript $(L)$ denotes low-degree joint masking. This prioritizes the masking of low-degree joints, producing the masked view $x_j^\theta$.

### 3.2.2 Temporal Masking (Motion-Based Frame Selection)

To select informative or subtle motion patterns over time, we compute motion scores across frames. First, for each frame $t$, we define motion magnitude as the average joint displacement:

$$m(t) = \|x_{t+1} - x_t\|, \quad x_t \in \mathbb{R}^{C \times V}. \tag{3}$$

We then compute motion scores by averaging over all channels and joints:

$$a_t = \frac{1}{C \cdot V} \sum_{c=1}^{C} \sum_{v=1}^{V} m_c^v(t). \tag{4}$$

**High-Motion Temporal Masking (HMTM).** To mask high-motion frames, we select the top-$k$ frames with the largest motion scores $a_t$, denoted as $\text{TopK}(a, k)$, where TopK returns the indices of the $k$ largest elements in $a$. The corresponding frames in $\hat{x}$ are masked to produce $x_m^\phi = \text{HMTM}(\hat{x}, \text{TopK}(a, k))$.

**Low-Motion Temporal Masking (LMTM).** Conversely, to encourage learning from low-motion frames, we mask the bottom-$k$ frames with the smallest motion scores, denoted as $\text{BottomK}(a, k)$, where BottomK returns the indices of the $k$ smallest elements in $a$. This yields the masked view $x_m^\theta = \text{LMTM}(\hat{x}, \text{BottomK}(a, k))$.

To this end, we obtain two asymmetric augmented views: $(x_j^\theta, x_m^\theta)$, and $(x_j^\phi, x_m^\phi)$ along with the shared unmasked anchor input $x$. These views serve as inputs to the corresponding anchor, spatial, and temporal stream during pretraining.

### 3.3 Pretraining $f_\theta$ and $f_\phi$

As discussed in Section 1, we observed that a correlation exist between the degree of a joint and its corresponding motion. We leverage this insight by pretraining two ST-GCN-based encoders, $f_\theta$ and $f_\phi$, with asymmetric spatio-temporal masking strategies. Each encoder processes three parallel streams: the **anchor stream** encodes the unmasked input $x$, the **spatial stream** encodes the joint-masked view $x_j^k$, and the **temporal stream** encodes the motion-masked view $x_m^k$, where $k \in \{\theta, \phi\}$. The resulting feature embeddings from these streams are denoted as $\mathbf{h}^k$, $\mathbf{h}_j^k$, and $\mathbf{h}_m^k$, respectively, and are fed to a share Barlow Twins projector head $p(\cdot)$ to obtain projected embeddings $\mathbf{Z}^k$, $\mathbf{Z}_j^k$, and $\mathbf{Z}_m^k$.

**Loss Function.** To align the anchor embedding with each masked stream while minimizing redundancy, we use the Barlow Twins loss function (Zbontar et al., 2021). For a pair of projected embeddings $\mathbf{z}, \mathbf{z}' \in \mathbb{R}^{B \times D}$, the cross-correlation matrix $\mathbf{C} \in \mathbb{R}^{D \times D}$ is computed as:

$$\mathbf{C}_{ij} = \frac{\sum_{b=1}^{B} \mathbf{z}_{b,i} \cdot \mathbf{z}'_{b,j}}{\sqrt{\sum_{b=1}^{B} \mathbf{z}_{b,i}^2} \cdot \sqrt{\sum_{b=1}^{B} \left(\mathbf{z}'_{b,j}\right)^2}}, \tag{5}$$

where $B$ is the batch size, $D$ is the feature dimension, and $i, j \in \{1, \ldots, D\}$ denote the feature indices in the projected embedding vectors $\mathbf{z}$ and $\mathbf{z}'$, respectively.

To align the anchor stream embedding $\mathbf{Z}^k$ with the spatially masked stream $\mathbf{Z}_j^k$ for encoder $f_k$ (where $k \in \{\theta, \phi\}$), we compute the Barlow Twins loss $\mathcal{L}_1^k$ as:

$$\mathcal{L}_1^k = \sum_{i=1}^{D} \left(1 - \mathbf{C}_{ii}'^k\right)^2 + \lambda \sum_{i=1}^{D} \sum_{\substack{j=1 \\ j \neq i}}^{D} \left(\mathbf{C}_{ij}'^k\right)^2, \tag{6}$$

where $\mathbf{C}'^k \in \mathbb{R}^{D \times D}$ is the cross-correlation matrix between projected anchor embeddings $\mathbf{Z}^k$ and spatial embeddings $\mathbf{Z}_j^k$, and $\lambda$ is a regularization coefficient.

Similarly, to align the anchor embedding $\mathbf{Z}^k$ with the temporally masked stream $\mathbf{Z}_m^k$, we compute the loss $\mathcal{L}_2^k$ as:

$$\mathcal{L}_2^k = \sum_{i=1}^{D} \left(1 - \hat{\mathbf{C}}_{ii}^k\right)^2 + \lambda \sum_{i=1}^{D} \sum_{\substack{j=1 \\ j \neq i}}^{D} \left(\hat{\mathbf{C}}_{ij}^k\right)^2, \tag{7}$$

where $\hat{\mathbf{C}}^k \in \mathbb{R}^{D \times D}$ is the cross-correlation matrix between $\mathbf{Z}^k$ and $\mathbf{Z}_m^k$. The total loss for encoder $f_k$ is then the sum of the two alignment losses:

$$\mathcal{L}_{\text{total}}^k = \mathcal{L}_1^k + \mathcal{L}_2^k, \tag{8}$$

and the final pretraining objective across both encoders becomes:

$$\mathcal{L}_{\text{ASMa}} = \mathcal{L}_{\text{total}}^\theta + \mathcal{L}_{\text{total}}^\phi. \tag{9}$$

This objective encourages both encoders to learn diverse yet complementary representations aligned across masked and unmasked spatio-temporal views.

### 3.4 Feature Alignment

For downstream evaluation, we align and integrate the diverse representations learned by $f_\theta$ and $f_\phi$ using a bi-directional cross-attention mechanism. Let $\mathbf{h}^\theta \in \mathbb{R}^{B \times N \times D}$ and $\mathbf{h}^\phi \in \mathbb{R}^{B \times N \times D}$ denote the feature embeddings extracted from $f_\theta$ and $f_\phi$, respectively, where $B$ is the batch size, $N$ is the sequence length (frames), and $D$ is the embedding dimension.

To align $\mathbf{h}^\theta$ and $\mathbf{h}^\phi$, we apply Multi-Head Attention (MHA) in a bi-directional manner: $f_\theta$ attends to $f_\phi$, and vice versa. The attention mechanism follows the standard scaled dot-product attention (Vaswani, 2017):

$$\text{Attention}(Q, K, V) = \text{Softmax}\left(\frac{QK^\top}{\sqrt{d_k}}\right)V, \tag{10}$$

where $Q$, $K$, and $V$ are the query, key, and value matrices, and $d_k$ is the dimension of the keys. Specifically, we compute two attention flows where, $f_\theta$ attends to $f_\phi$ by setting $Q = \mathbf{h}^\theta$, $K = V = \mathbf{h}^\phi$, producing $\mathbf{h}^\theta_{\text{aligned}}$ and $f_\phi$ attends to $f_\theta$ by setting $Q = \mathbf{h}^\phi$, $K = V = \mathbf{h}^\theta$, producing $\mathbf{h}^\phi_{\text{aligned}}$.

We then fuse the outputs as shown in Eq. 11:

$$\mathbf{h}_{\text{aligned}} = \mathbf{h}^\theta_{\text{aligned}} + \mathbf{h}^\phi_{\text{aligned}}. \tag{11}$$

The resulting unified representation $\mathbf{h}_{\text{aligned}}$ is passed through a linear classification head:

$$\hat{\mathbf{y}} = \mathbf{W}_{\text{cls}} \cdot \mathbf{h}_{\text{aligned}}, \tag{12}$$

where $\mathbf{W}_{\text{cls}} \in \mathbb{R}^{D \times C}$ is the classification weight matrix, and $C$ is the number of action classes. The full feature alignment module, including the MHA blocks and classifier head, is trained end-to-end using standard cross-entropy loss. We use 4 heads in MHA in this module.

### 3.5 Knowledge Distillation

To enable efficient deployment, we distill the unified representation learned by the feature alignment module into a lightweight student encoder $f_s$. This step reduces model size and inference cost while aiming to retain most of the performance achieved by the full ASMa framework. We design a teacher-student distillation framework, where the teacher is defined as the full model comprising $f_\theta$, $f_\phi$, and the feature alignment module. During training, the teacher model is kept frozen to prevent collapse and to provide stable supervision. Both the teacher and the student $f_s$ take the same input skeleton sequence $x$ and produce logits $\mathbf{z}_k$, and $\mathbf{z}_s$ respectively. To facilitate effective distillation, we apply temperature scaling to the teacher logits. The softened distribution over classes is computed as:

$$\hat{y}_i^T = \frac{e^{z_{k,i}/\tau}}{\sum_{j=1}^C e^{z_{k,j}/\tau}}, \tag{13}$$

where $z_{k,i}$ is the $i$-th component of the teacher logits $\mathbf{z}_k$, $\tau$ is the temperature parameter, $C$ is the total number of classes, and $i$ indexes the class dimension. Temperature scaling helps preserve class-level similarity by smoothing the distribution. The student output is similarly converted into a probability distribution $\hat{\mathbf{y}}^S$ using softmax. We minimize the Kullback-Leibler (KL) divergence between the student and teacher distributions:

$$\mathcal{L}_{\text{kd}} = \mathcal{L}_{\text{KL}}(\hat{\mathbf{y}}^S \| \hat{\mathbf{y}}^T). \tag{14}$$

## 4 Datasets and Implementation

**Datasets.** We evaluate ASMa on three standard benchmarks: NTU RGB+D 60 (Shahroudy et al., 2016), NTU RGB+D 120 (Liu et al., 2019), and PKU-MMD (Liu et al., 2020a). NTU-60 consists of 56,578 skeleton sequences from 60 actions across 40 subjects; we use the standard cross-subject (xsub) and cross-view (xview) splits. NTU-120 extends this with 113,945 sequences across 120 actions and 106 subjects, evaluated under cross-subject (xsub) and cross-setup (xset) protocols. PKU-MMD contains 28,443 sequences (Part I and II combined) covering 51 actions. We use Part I for pretraining and both parts for transfer evaluation. PKU-MMD presents significant challenges due to viewpoint variation and skeleton noise in Part II.

**Implementation Details.** For all experiments, we use the ST-GCN (Yan et al., 2018) backbone with 16 hidden channels and train with Adam optimizer and CosineAnnealing scheduler for 150 epochs. Batch size

is set to 128. We pretrain $f_\theta$ and $f_\phi$ on the xsub splits of NTU-60 and NTU-120, and on Part I of PKU-MMD. We mask 9 joints and 10 frames during pretraining, based on the ablation shown in Appendix A.2. Each encoder outputs a 256-dimensional vector, projected to a 6144-dimensional space. The loss trade-off parameter $\lambda$ is set to $2 \times 10^{-4}$, and weight decay is 1e-5 with 10 warmup epochs.

**Downstream Evaluation.** We evaluate pretrained models on action classification via (i) Linear Evaluation and (ii) Fine-tuning. In linear evaluation, we freeze $f_\theta$ and $f_\phi$ and train only the feature alignment module and classification head for 150 epochs with a learning rate of 1e-3. We also report linear-probing results from individual encoders. In fine-tuning, we train each encoder end-to-end with a linear head for 150 epochs at a 5e-3 learning rate. For the combined fine-tuning, we attach our feature alignment module on top of fine-tuned $f_\theta$ and $f_\phi$ and train it for 50 epochs with a learning rate of 1e-4.

**Knowledge Distillation.** The student model $f_s$ consists of 5 ST-GCN layers with spatial kernel size 3 and temporal kernel size 9. It is trained using KL divergence loss (without ground-truth supervision) for 150 epochs at a learning rate of 1e-2.

## 5 Experimental Validation

Our experimental validation addresses the following questions:

- **RQ1 Action Classification & Efficiency:** How do (three-stream) 3s-ASMa and 3s-ASMa-Distill compare with supervised and self-supervised baselines under linear evaluation and fine-tuning, and what accuracy–efficiency trade-offs do they offer (including on-device performance)?

- **RQ2 Ablations Study:** What are the individual contributions of the two encoders ($f_\phi$, $f_\theta$), and how much does the feature alignment module help? How do spatial–temporal masking pairings behave, and does asymmetric masking outperform symmetric/random choices?

- **RQ3 Distillation Sensitivity:** How sensitive is ASMa-Distill to distillation hyperparameters, particularly temperature $\tau$ and the student's depth (number of ST-GCN layers)?

- **RQ4 Linear Probed vs Finetuned KD:** How does the linear-probed vs. fine-tuned—affect student performance and the quality of learned representations?

- **RQ5 Transfer Learning Evaluation:** How well do ASMa's representations transfer to noisy datasets?

### 5.1 RQ1: Action Classification and Efficiency

In Table 1, we report the performance of both our proposed 3s-ASMa and its distilled variant, 3s-ASMa-Distill, alongside state-of-the-art supervised and self-supervised methods. The table presents linear evaluation (Lin.) and fine-tuned (FT.) top-1 accuracy across NTU RGB+D 60, NTU RGB+D 120, and PKU-MMD, together with efficiency metrics including parameter count and FLOPs. The prefix 3-Stream (3s) denotes that results are obtained by combining the joint, bone, and motion streams.

**Discussion.** From Table 1, several key observations emerge. First, 3s-ASMa achieves consistent improvements over prior self-supervised learning approaches, surpassing them by up to 4–6% in fine-tuning accuracy across NTU-60, NTU-120, and PKU-MMD. These gains highlight the importance of our asymmetric masking strategy for learning richer skeleton representations and the role of the feature alignment module in enhancing downstream generalization. Second, while ASMa establishes a new state-of-the-art benchmark, we recognize that deploying such models in edge scenarios is challenging due to their size and computational overhead. To address this, we introduce 3s-ASMa-Distill, a lightweight student model distilled from ASMa. Despite having only 0.54M parameters and 1.26G FLOPs, the distilled model not only preserves strong accuracy but also outperforms all previous baselines on NTU-60 and NTU-120. These results demonstrate that our framework delivers both state-of-the-art accuracy and practical efficiency, making it suitable for real-world deployment on resource-constrained devices.

Table 1: Comparison of action recognition results on NTU RGB+D 60, NTU RGB+D 120, and PKU-MMD with efficiency metrics. Results are reported for linear evaluation (Lin.) and fine-tuning (FT.). Params (millions) and FLOPs (G) are included where available. Backbone (Bk.) identifiers: (1) ST-GCN, (2) GCN, (3) CNN, (4) DGCNN, (5) STTFormer, (6) Transformer, (7) GCN+Transformer.

| Method | Bk. | Params | FLOPs | NTU 60 | | | | NTU 120 | | | | PKU-MMD | | | |
| | | | | xsub | | xview | | xsub | | xset | | Part I | | Part II | |
| | | | | Lin. | FT. | Lin. | FT. | Lin. | FT. | Lin. | FT. | Lin. | FT. | Lin. | FT. |
| **Fully-Supervised:** | | | | | | | | | | | | | | | |
| ST-GCN (Yan et al., 2018) | 1 | – | – | – | 81.5 | – | 88.3 | – | – | – | – | – | – | – | – |
| 2s-AGCN (Shi et al., 2019) | 2 | – | – | – | 88.5 | – | 95.1 | – | 82.9 | – | 84.9 | – | – | – | – |
| Shift-GCN (Cheng et al., 2020) | 2 | – | – | – | 90.7 | – | 96.5 | – | 85.9 | – | 87.6 | – | – | – | – |
| MS-G3D (Liu et al., 2020b) | 2 | – | – | – | 91.5 | – | 96.2 | – | 86.9 | – | 88.4 | – | – | – | – |
| PYSKL (Duan et al., 2022) | 3 | – | – | – | 93.7 | – | 96.6 | – | 86.0 | – | 89.6 | – | – | – | – |
| CTR-GCN (Chen et al., 2021) | 2 | – | – | – | 92.4 | – | 96.8 | – | 88.9 | – | 90.6 | – | – | – | – |
| BlockGCN Zhou et al. (2024) | 2 | – | – | – | 93.1 | – | 97.0 | – | 90.3 | – | 91.5 | – | – | – | – |
| HiOD Chang et al. (2025) | 2 | – | – | – | **93.8** | – | **97.6** | – | **90.4** | – | **91.9** | – | – | | |
| **Self-Supervised:** | | | | | | | | | | | | | | | |
| 3s-SkeletonBT (Zhou et al., 2023) | 1 | – | – | 69.3 | – | 72.2 | – | 56.3 | – | 56.8 | – | 84.8 | – | 44.1 | – |
| 3s-SkeletonCLR (Li et al., 2021) | 1 | – | – | 75.0 | – | 79.8 | – | 60.7 | – | 62.6 | – | – | – | – | – |
| 3s-ST-GCN (Yan et al., 2018) | 1 | – | – | – | 85.2 | – | 91.4 | – | 77.2 | – | 77.1 | – | – | – | – |
| 3s-CrosSCLR (Li et al., 2021) | 1 | 26.6 | 17.28 | 77.8 | 86.2 | 83.4 | 92.5 | 67.9 | 80.5 | 67.1 | 80.4 | 84.9 | – | 21.2 | – |
| SkeletonMAE (Wu et al., 2023) | 5 | – | – | – | 86.6 | – | 92.9 | – | 76.8 | – | 79.1 | – | – | – | – |
| 3s-AimCLR (Guo et al., 2022) | 1 | 2.85 | 3.45 | 78.9 | 86.9 | 83.8 | 92.8 | 68.2 | 80.1 | 68.8 | 80.9 | 87.8 | – | 38.5 | – |
| 3s-colorization (Yang et al., 2021) | 4 | – | – | 75.2 | 88.0 | 83.1 | 94.9 | – | – | – | – | – | – | – | – |
| 3s-PSTL (Zhou et al., 2023) | 1 | 2.85 | 3.45 | 79.1 | 87.1 | 83.8 | 93.9 | 69.2 | 81.3 | 70.3 | 82.6 | 89.2 | – | 52.3 | 69.1 |
| SCD-Net (Wu et al., 2024) | 7 | – | – | 86.6 | – | 91.7 | – | 76.9 | – | 80.1 | – | 91.9 | – | 54.0 | – |
| IGM (Lin et al., 2025a) | 6 | – | – | – | 86.2 | – | 91.2 | – | 80.0 | – | 81.4 | – | – | – | – |
| ActCLR (Lin et al., 2025b) | 2 | – | – | – | 89.0 | – | 94.2 | – | 82.2 | – | 84.8 | 91.6 | – | 62.1 | – |
| STJD-CL (Gunasekara et al., 2025) | 2 | – | – | – | 89.3 | – | 94.8 | – | 83.5 | – | 86.8 | 93.2 | – | 55.3 | – |
| **3s-ASMa** | 1 | 6.3 | 4.5 | **87.3** | **92.0** | **91.9** | **96.8** | **80.1** | **87.9** | **81.0** | **88.8** | **94.5** | – | **68.9** | **76.8** |
| | | | | ↑0.7 | ↑2.7 | ↑0.2 | ↑1.9 | ↑3.2 | ↑4.3 | ↑0.9 | ↑2.0 | ↑1.3 | – | ↑14.9 | ↑7.7 |
| **3s-ASMa-Distill** | 1 | **0.54** | **1.26** | – | **91.7** | – | **95.9** | – | **86.9** | – | **88.3** | – | – | – | **75.8** |
| | | ↓91.4% | ↓72% | – | ↑2.4 | – | ↑1.0 | – | ↑3.4 | – | ↑1.5 | – | – | – | ↑6.7 |

Table 2: Accuracy and inference efficiency comparison between 3s-ASMa and 3s-ASMa-Distill on Raspberry Pi (2GB RAM, CPU only).

| Model | NTU60 (FT) | | NTU120 (FT) | | Params(M) | Size(MB) | FLOPs(G) | Time(ms) | Mem(MB) | FPS |
| | xsub | xview | xsub | xset | | | | | | |
| 3s-ASMa (Teacher) | **92.0%** | **96.8%** | **87.9%** | **88.7%** | 6.3 | 25.5 | 4.5 | 59.16 | 314.47 | 16.90 |
| 3s-ASMa-Distill | 91.7% | 95.9% | 86.9% | 88.3% | **0.54** | **2.22** | **1.26** | **21.38** | **173.28** | **46.52** |
| Performance/Efficency | ↓0.3% | ↓0.9% | ↓1.0% | ↓0.4% | ↓91.4% | ↓91.2% | ↓72% | ↓63.8% | ↓44.8% | ↑63.6% |

**On Edge Performance:** To assess practical applicability, we benchmark ASMa-Distill on a Raspberry Pi 4B (2GB RAM, dual-core CPU). As shown in Table 2, the student model achieves only 0.65% average performance drop while reducing parameters by 91%, disk footprint by 91.2%, and FLOPs by 72%. It runs $3\times$ faster (21.38ms vs. 59.16ms), uses 45% less memory, and reaches 46.52 FPS, demonstrating feasibility for real-time edge deployment.

## 5.2 RQ2: Ablation Study

We perform an ablation of the two encoders $f_\phi$ and $f_\theta$ and the feature-alignment module. Both encoders use the same ST-GCN backbone and training setup; they differ only in their masking strategy. Thus, their comparison reflects the effect of different spatio-temporal masking configurations rather than architecture. As shown in Table 3, each encoder captures complementary motion cues, and their combination improves accuracy by 1–2% across datasets.

Table 3: Ablation study of individual encoders and the feature alignment (FA) module.

| Model | NTU60 | | NTU120 | | PKU-MMD | |
| | xsub | xview | xsub | xset | Part I | Part II |
| $f_\phi$ only | 83.8 | 88.2 | 75.1 | 75.9 | 91.0 | 64.3 |
| $f_\theta$ only | 84.1 | 89.1 | 76.0 | 76.8 | 91.7 | 65.1 |
| $f_\phi + f_\theta$ (w/o FA) | 85.9 | 90.5 | 78.3 | 79.0 | 93.6 | 68.2 |
| $f_\phi + f_\theta$ (w/ FA) | **87.3** | **91.9** | **80.1** | **81.0** | **94.5** | **68.9** |

Removing the feature-alignment module and averaging encoder outputs lowers accuracy by up to 2%, confirming the benefit of adaptive fusion.

To study how spatial and temporal masking interact, we ablate ASMa on the joint stream using all pairwise combinations of Spatio-temporal masking as shown in Figure 3.

**Discussion.** Across both NTU60 splits (xsub/xview), the best performance is obtained when masking low-degree joints with high-motion frames; the second-best comes from the complementary pairing of high-degree joints with low-motion frames (highlighted in the figure). In contrast,

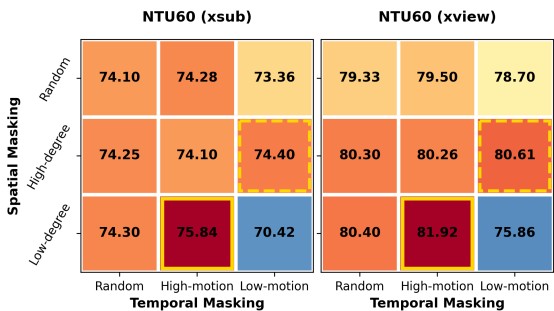

Figure 3: Ablation on different pairs of masking strategies. Refer to Appendix B.1 for other benchmarks.

symmetric pairings (high+high or low+low) and random masking are consistently weaker. These results align with our observation that high-degree joints typically move less while low-degree joints exhibit greater motion (see Fig. 1) and support our design choice of asymmetric spatio-temporal masking in Fig. 2(b). By cross-pairing spatial and temporal difficulty, the pretext task exposes complementary cues and encourages more discriminative representations, providing direct empirical evidence for our claim.

To evaluate the architectural flexibility of ASMa, we conduct experiments using three popular backbone networks: Graph Isomorphism Network (GIN), Dynamic Graph CNN (DGCNN), and Spatio-Temporal GCN (ST-GCN), as shown in Table 4. While ASMa demonstrates consistent gains across all three backbones under both linear evaluation and fine-tuning

Table 4: Performance of ASMa across different backbones (Bk).

| Bk | NTU60 | | | | NTU120 | | | |
|---|---|---|---|---|---|---|---|---|
| | xsub | | xview | | xsub | | xset | |
| | Lin. | FT. | Lin. | FT. | Lin. | FT. | Lin. | FT. |
| GIN | 75.23 | 82.43 | 81.34 | 91.05 | 64.47 | 75.16 | 65.87 | 77.12 |
| DGCNN | 75.10 | 83.29 | 81.48 | 91.30 | 65.42 | 75.62 | 66.94 | 77.29 |
| ST-GCN | **75.84** | **84.08** | **81.92** | **91.53** | **65.73** | **76.39** | **67.08** | **77.58** |

protocols, ST-GCN achieves the highest accuracy on all splits. This suggests that although ASMa generalizes well to different backbone designs, ST-GCN is particularly effective in capturing spatio-temporal dependencies in skeleton sequences.

### 5.3 RQ3: Distillation sensitivity

In this section, we discuss the effect of temperature parameter $\tau$ of Eq. 13 and the number of ST-GCN layers on the performance of our compact model $f_s$. First, to study the effect of temperature-scaled distillation, we keep the number of ST-GCN layers fixed to 5 in $f_s$ and vary the value of $\tau$.

**Discussion.** In general, as shown in Figure 4 (Left), we observe a similar performance pattern where too low or high value of $\tau$ results in a performance drop across all benchmarks. However, the optimal value of $\tau$ depends on the class distribution of the datasets. Specifically, for datasets with fewer classes, such as NTU60 (60 classes), our intuition suggests that the output distributions are inherently sharper and require higher $\tau$ values (e.g., 8,9) for effective softening of logits and better distillation. For example, NTU60-xview achieves its highest performance (90.75%) at $\tau = 9$. In contrast, datasets with more classes, such as NTU120 (120 classes), naturally produce smoother output logits. Excessive softening with a higher value of $\tau$ can blur critical inter-class distinctions, leading to suboptimal performance. Notably, NTU120-xsub achieves 77.42% accuracy at $\tau = 6$, outperforming higher values. Next, we analyze the effect of increasing and decreasing number of ST-GCN layers in $f_s$. As shown in Figure 4 (Right), there is a consistent performance improvement with increasing number of layers. However, increasing the number of layers beyond 5 results in marginal improvements on all benchmarks. Specifically, NTU60-xsub improves by only 0.49%, and NTU60-xview gains 0.28% when moving from 5 to 7 layers. However, reducing the number of layers to 3 or 2 leads to a noticeable performance drop across all datasets. To this end, we find that 5 ST-GCN layers in $f_s$ show

a reasonable trade-off between performance and compression. A detailed comparison between feature-level and logit-level distillation, including implementation and quantitative results, is presented in Appendix C.

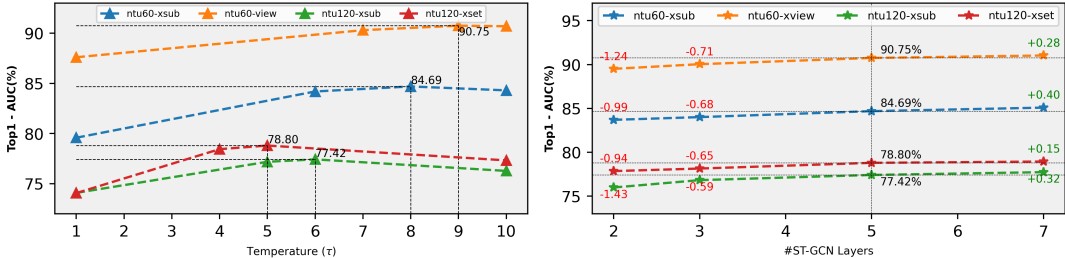

Figure 4: **Left:** We keep the number of ST-GCN layer in $f_s$ to 5 and vary $\tau$ to test temperature sensitivity of distillation. **Right:** We vary the number number of ST-GCN layers in $f_s$ to test compression sensitivity.

### 5.4 RQ4: Linear Probed vs Finetuned KD

While temperature scaling is a common approach to soften the logits for better knowledge transfer, it is inherently a heuristic method. This raises an interesting question: ***What if we distill from a linear-probed teacher, which naturally produces softer logits due to its constrained learning setup?***

To find the answer, we conduct experiments to compare the performance of students when distilled from a linear-probed (LP) teacher vs a fine-tuned (FT) teacher. Interestingly, as shown in Table 5, the student distilled from the LP teacher consistently outperforms its teacher on all datasets, with an average performance gain of 5.2%.

**Discussion.** To understand why $f_s$ performs better than the teacher when distilled from LP model, we analyze the t-SNE embedding projection in Figure 5. We choose all data samples from 9 random classes of NTU60-xsub (test set) for clear visualization. As shown in Figure 5, LP-distilled student (bottom right) demonstrates more compact and well-separated clusters compared to its teacher (bottom left). In contrast, the FT-distilled student (top right) shows slightly dispersed clusters compared to its FT teacher (top left). Intuitively speaking, since the student has nearly 12 times fewer parameters than the teacher, we observe an aggregated effect of representation learned by the teacher which leads to more refined and compact clusters in the LP-distilled case. However, in the FT setting, since the decision boundaries are more fine-grained, the student struggles to retain the performance of the teacher due to lower capacity. Note that although distillation from

Table 5: Comparison of distillation from linear-probed vs fine-tuned ASMa. Refer to Appendix A.3 for other streams.

| Stream | Variant | NTU60 | | | | NTU120 | | | |
|---|---|---|---|---|---|---|---|---|---|
| | | xsub | | xview | | xsub | | xset | |
| | | Lin. | FT. | Lin. | FT. | Lin. | FT. | Lin. | FT. |
| **Bone (B)** | ASMa (T) | 75.8 | **85.5** | 81.1 | **92.6** | 66.1 | **79.2** | 66.9 | **80.2** |
| | ASMa (S) | **80.3** | 84.6 | **85.5** | 90.7 | **72.0** | 77.4 | **73.2** | 78.8 |
| | ↑/↓ | ↑4.5 | ↓0.9 | ↑4.4 | ↓1.9 | ↑5.9 | ↓1.8 | ↑6.3 | ↓1.4 |

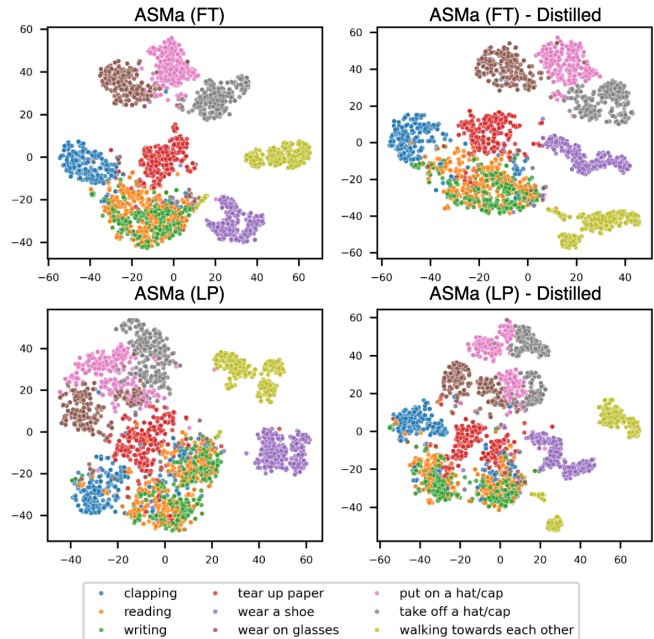

Figure 5: t-SNE embedding projections of 9 randomly selected classes from NTU60-xsub (test set). Refer to Appendix A.4 for other classes.

a fine-tuned teacher shows a better performance than the linear-probed teacher, it comes with the cost of finding the perfect value of $\tau$ for better distillation. Therefore, in scenarios where training multiple models is not viable due to time constraints, distillation from linear-probed teachers offers a viable solution in case of skeleton representation learning with a reasonable trade-off on performance.

### 5.5 RQ5: Transfer Learning Evaluation

Table 6 shows that ASMa achieves notable improvements in transfer learning, outperforming baselines by 4-6% on PKU-Part-II, a dataset known for its high noise levels.

**Discussion:** This gain highlights the superior generalization ability of our model with its asymmetric masking strategy and feature alignment, which enforces learning from both high-degree joints with low motion and low-degree joints with high motion. By capturing a broader range of motion dynamics, ASMa learns more comprehensive representations, which makes it better suited for challenging and noisy datasets.

Table 6: Transfer learning performance (FT). Models are pre-trained on NTU60-xsub, NTU120-xsub, or PKU-Part-I and transferred to PKU-Part-II. * indicates results reproduced using official code of (Zhou et al., 2023).

| Method | Transfer to PKU-Part-II | | |
|---|---|---|---|
| | **NTU60** | **NTU120** | **PKU** |
| LongTGAN (Zheng et al., 2018) | 44.8 | – | 43.6 |
| MS2L (Lin et al., 2020) | 45.8 | – | 44.1 |
| ISC (Thoker et al., 2021) | 51.1 | 52.3 | 45.1 |
| CMD (Mao et al., 2022) | 56.0 | 57.0 | – |
| SkeletonMAE (Yan et al., 2023) | 58.4 | 61.0 | 62.5 |
| S-JEPA (Abdelfattah & Alahi, 2024) | 71.4 | 74.2 | 70.9 |
| 3s-PSTL* (Zhou et al., 2023) | 72.4 | 70.1 | 69.1 |
| **3s-ASMa** | **77.2** | **77.0** | **76.8** |
| | ↑ 4.8 | ↑ 2.8 | ↑ 5.9 |

## 6 Conclusion

We proposed Asymmetric Spatio-Temporal Skeleton Action Representation Learning (ASMa), a self-supervised learning framework that enhances skeleton-based action recognition by leveraging asymmetric masking in spatial and temporal streams. Our feature alignment module effectively integrates diverse representations, and knowledge distillation enables a lightweight model suitable for resource-constrained environments. Experiments on NTU RGB+D 60, NTU RGB+D 120, and PKU-MMD show that ASMa surpasses previous self-supervised methods and achieves competitive performance with supervised models. Notably, distillation from a linear-probed teacher yields a more compact and generalizable student model. Future work includes extending ASMa to multi-modal learning and exploring asymmetric spatio-temporal masking to improve the generalization of SSL frameworks. While ASMa achieves strong performance across benchmarks, we acknowledge scope considerations for future extensions. First, like other skeleton-based methods, performance depends on pose estimation quality; however, our strong transfer results on PKU-MMD Part II (Table 6), a dataset with significant noise and viewpoint variation, suggest that the asymmetric masking strategy helps mitigate noise effects by capturing complementary motion dynamics. Second, ASMa is designed for single-person action recognition, consistent with standard benchmarks, and does not explicitly model multi-person interactions or occlusions. Extending to multi-person scenarios and further improving noise handling remain promising directions for future work.

### Broader Impact Statement

Skeleton-based action recognition offers privacy advantages over RGB video by reducing personally identifiable information. However, like all surveillance technologies, it could be misused for non-consensual tracking. Additionally, performance may vary across populations due to dataset biases; practitioners should evaluate fairness across diverse user groups. We encourage practitioners to obtain informed consent, comply with local privacy regulations (e.g., GDPR), and deploy our methods only in approved applications. The datasets used in this work (NTU RGB+D, PKU-MMD) are standard public benchmarks collected with participant consent and widely adopted in the research community.

**Acknowledgments**

This work was undertaken thanks in part to funding from the Connected Minds program, supported by Canada First Research Excellence Fund, grant #CFREF-2022-00010. This work was also supported by NSERC Discovery RGPIN-2018-05550.

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

# Appendix

## A  Extended Experiments and Analysis

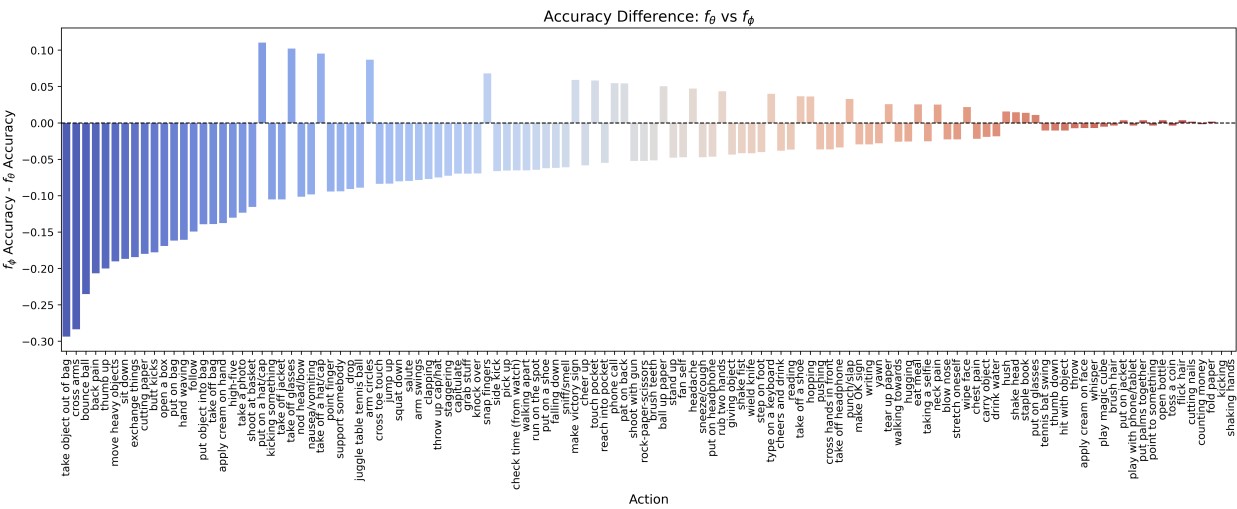

Figure 6: Accuracy difference per action class between $f_\theta$ and $f_\phi$. Positive values indicate actions where $f_\phi$ outperforms $f_\theta$, while negative values indicate better performance of $f_\theta$.

### A.1  Encoder-wise Performance Analysis

Figure 6 presents a class-wise comparison of the accuracy difference between the two encoders, $f_\theta$ and $f_\phi$. The values indicate the relative advantage of one encoder over the other for specific action categories. Positive values show actions where $f_\phi$ achieves higher accuracy, while negative values indicate better performance of $f_\theta$. We observe that, while $f_\theta$ generally outperforms $f_\phi$, there are several action categories where $f_\phi$ exhibits superior accuracy. This suggests that $f_\phi$ captures certain nuances in the data that $f_\theta$ may overlook. The complementary nature of both encoders highlights the importance of integrating diverse perspectives in skeleton-based representation learning. Specifically, the presence of actions where $f_\phi$ excels indicates that relying solely on one masking strategy may result in incomplete feature learning. This reinforces the necessity of aligning the representations learned from both encoders to achieve a more comprehensive understanding of human actions.

### A.2  Effect of Masking Ratio.

We perform an ablation study on the NTU RGB+D 60 dataset to assess the sensitivity of ASMa to the number of masked joints ($n$) and frames ($k$) during pretraining. As shown in Table 7, we observe that performance peaks when $n = 9$ joints and $k = 10$ frames are masked. Increasing the masking beyond these values degrades performance, likely due to excessive information loss, while lower masking levels reduce the diversity of views presented during training. These findings empirically support our choice of masking 9 joints and 10 frames during pretraining.

### A.3  Distillation Performance

In the main paper, we analyzed the impact of distillation from a linear-probed (LP) teacher versus a fine-tuned (FT) teacher, focusing solely on the Bone stream(Table 5). We observed that distilling from the LP teacher led to a more generalizable student model, outperforming its own teacher while maintaining compactness. To extend this analysis, Table 8 presents the distillation results across all streams, including Joint and Motion, in both linear evaluation and fine-tuning settings.

Table 7: Ablation study on the NTU RGB+D 60 dataset evaluating the impact of varying the number of masked joints ($n$) and masked frames ($k$) during pretraining. All results are reported using linear probe accuracy (%) under the 3s-ASMa setting.

| Setting | Mask Count | 3s-ASMa Accuracy (%) |
|---|---|---|
| | 7 | 86.1 |
| Masked Joints ($n$), $k = 10$ fixed | 8 | 87.1 |
| | 9 | **87.3** |
| | 10 | 86.5 |
| | 7 | 85.2 |
| | 8 | 86.1 |
| Masked Frames ($k$), $n = 9$ fixed | 9 | 86.8 |
| | 10 | **87.3** |
| | 11 | 87.0 |

Table 8: Accuracy of Distillation from Linear probed vs Finetuned ASMa on all streams.

| Stream | Variant | NTU-60 (%) | | | | NTU-120 (%) | | | |
|---|---|---|---|---|---|---|---|---|---|
| | | xsub | | xview | | xsub | | xset | |
| | | Lin. | FT. | Lin. | FT. | Lin. | FT. | Lin. | FT. |
| **Joint (J)** | ASMa | 76.5 | **84.9** | 83.8 | **92.3** | 66.4 | **77.1** | 67.5 | **78.3** |
| | ASMa-Distill | **80.5** | 84.1 | **86.8** | 90.5 | **71.4** | 76.0 | **73.2** | 76.8 |
| **Motion (M)** | ASMa | 69.5 | **82.9** | 74.7 | **90.4** | 58.7 | **74.0** | 59.7 | **74.8** |
| | ASMa-Distill | **75.5** | 81.8 | **80.0** | 88.3 | **67.5** | 73.3 | **65.9** | 73.7 |
| **Bone (B)** | ASMa | 75.8 | **85.5** | 81.1 | **92.6** | 66.1 | **79.2** | 66.9 | **80.2** |
| | ASMa-Distill | **80.3** | 84.6 | **85.5** | 90.7 | **72.0** | 77.4 | **73.2** | 78.8 |

The trends observed in the Bone stream persist across other streams as well. ASMa-Distill surpasses ASMa in the linear evaluation setting for all streams, reinforcing the hypothesis that a linear-probed teacher provides a more transferable representation. Notably, in the Joint stream, ASMa-Distill achieves a 4-6% gain in linear evaluation, further confirming that LP-trained teachers encode smoother decision boundaries that are beneficial for distillation. Similarly, for the Motion stream, the distilled model exhibits substantial improvements over its teacher in linear evaluation.

Although distillation from a fine-tuned model shows a better performance over distillation from the linear-probed model, it comes with the cost of finding the perfect value of $\tau$ for better distillation. Therefore, in scenarios where training multiple models is not viable due to time constraints, distillation from linear-probed teachers offers a viable solution with a reasonable trade-off on performance.

### A.4 Additional t-SNE Embedding Projections for Distillation Analysis

To further analyze the effect of distillation from a linear-probed (LP) versus fine-tuned (FT) teacher, we provide additional t-SNE embedding projections in Figures 7a and 7b. Each figure visualizes 10 randomly selected action classes from NTU60-xsub (test set), offering a broader perspective on the representation learned by different models.

Consistent with the observations in Figure 5 from the main paper, we see that the LP-distilled student exhibits more compact and well-separated clusters compared to its FT-distilled counterpart. This suggests that distillation from a linear-probed teacher, which inherently produces softer logits, helps the student

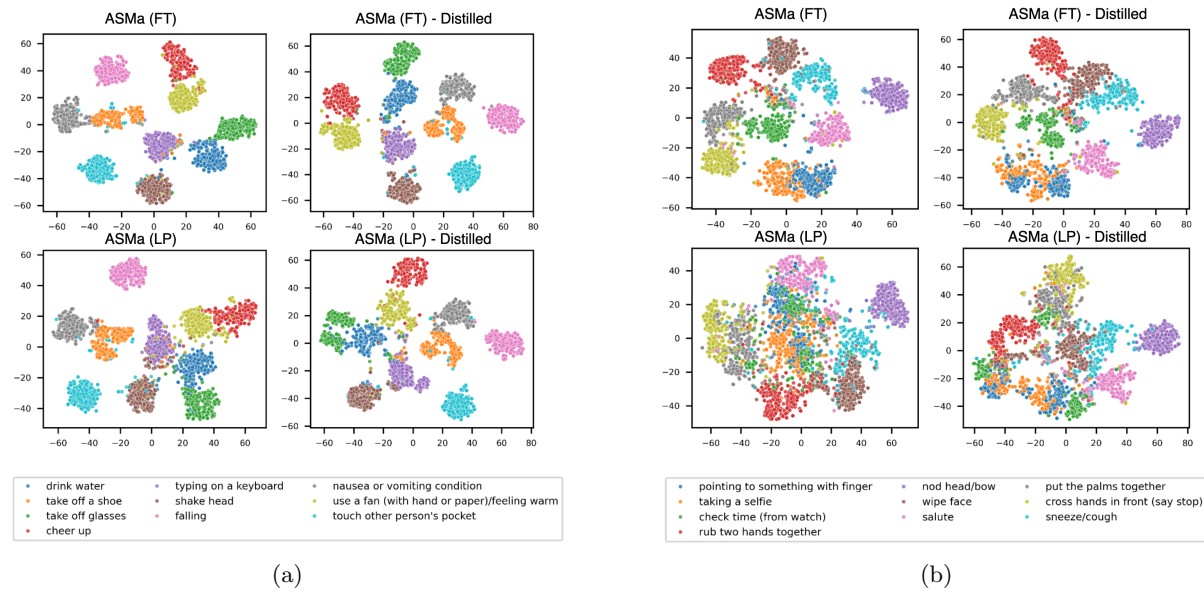

Figure 7: t-SNE embedding projections of 10 randomly selected classes from NTU60-xsub (test set). (a) Random set 1. (b) Random set 2.

network learn more generalizable and structured representations. On the other hand, the FT-distilled student struggles to retain the fine-grained decision boundaries of its teacher, leading to slightly more dispersed clusters.

# B   Ablation on Masking strategy and Prior Frameworks

## B.1   Combination of Masking Strategy

To further strengthen the analysis, we evaluate different spatial–temporal masking combinations across the NTU RGB+D 60, NTU RGB+D 120, and PKU-MMD benchmarks. All experiments use the same ST-GCN backbone and hyperparameters (Adam optimizer, 150 epochs, batch size 128, masking 9 joints and 10 frames). Only the masking configuration is varied. As shown in Table 9, the asymmetric combinations, masking low-degree joints with high-motion frames or high-degree joints with low-motion frames, consistently achieve the best results across all datasets. This emperically validates that the proposed asymmetric masking design generalizes well across benchmarks.

Table 9: Ablation study on different combination of spatial and temporal masking strategies. Linear evaluation accuracy (%) is reported for NTU RGB+D 60, NTU RGB+D 120, and PKU-MMD datasets.

| Masking Strategy | | NTU60 | | NTU120 | | PKU-MMD | |
|---|---|---|---|---|---|---|---|
| Spatial | Temporal | xsub | xview | xsub | xset | Part I | Part II |
| Random | Random | 74.2 | 79.3 | 63.4 | 64.25 | 82.92 | 47.26 |
| Low-Degree | Low-Motion | 70.42 | 75.86 | 62.34 | 63.21 | 81.24 | 45.15 |
| High-Degree | High-Motion | 74.1 | 80.26 | 64.05 | 64.96 | 84.70 | 47.93 |
| *High-Degree* | *Low-Motion* | *74.38* | *80.61* | *64.81* | *65.30* | *84.90* | *52.94* |
| **Low-Degree** | **High-Motion** | **75.84** | **81.92** | **66.73** | **67.80** | **87.74** | **54.35** |

## B.2 Generalization of ASMa as an Augmentation Strategy

To examine the generality of the proposed Asymmetric Spatio-Temporal Masking (ASMa) strategy, we retrained previous SSL frameworks by replacing only their original data augmentations with ASMa, while keeping all other components, training objectives, and hyperparameters identical to their official implementations. The evaluated methods include MS$^2$L, 3s-AimCLR, and 3s-CrossCLR. Training followed the configurations in their respective papers: MS$^2$L used multi-task learning with motion prediction, jigsaw puzzle, and contrastive losses (learning rate $1 \times 10^{-3}$, batch size 128, SGD, 120 epochs), while 3s-AimCLR and 3s-CrossCLR used temperature $\tau = 0.07$, momentum $m = 0.999$, memory bank size 65,536, and learning rate 0.1 with CosineAnnealing for 300 epochs. The original augmentations (e.g., Shear+Crop in AimCLR, Cross-View rotation in CrossCLR) were replaced by ASMa's asymmetric masking.

As shown in Table 10, consistent improvements (1.0–2.3%) across all baselines demonstrate that ASMa serves as a general augmentation mechanism enhancing representation learning regardless of the SSL objective. These results highlight ASMa's ability to expose more informative spatio-temporal dependencies than standard geometric augmentations.

Table 10: Performance of existing SSL methods equipped with the proposed ASMa augmentation. M$^2$SL* denotes results reproduced using official code. Values in parentheses indicate improvement over the baseline.

| Method | NTU60 (x-sub) | NTU60 (x-view) |
|---|---|---|
| M$^2$SL* | 84.6 | 91.3 |
| M$^2$SL* + **ASMa** | **86.0** (+1.4) | **93.6** (+2.3) |
| 3s-AimCLR | 86.9 | 92.8 |
| 3s-AimCLR + **ASMa** | **88.2** (+1.3) | **95.1** (+2.3) |
| 3s-CrossCLR | 86.2 | 92.5 |
| 3s-CrossCLR + **ASMa** | **87.2** (+1.0) | **94.0** (+1.5) |

## C Feature vs Logit Distillation

We empirically compare feature-level and logit-level distillation to examine their effect on performance and complexity. For feature-level distillation, an additional linear projection layer was added to the teacher to match the 128-D student representation. The model was trained with an L2-normalized cosine similarity loss for 150 epochs (learning rate $1 \times 10^{-3}$). As shown in Table 11, after fine-tuning, the student achieved lower accuracy than with logit-level distillation.

## D Statistical Significance Analysis

To evaluate the statistical significance and reproducibility of our results, we conducted 5 independent runs with different random seeds for all 3s-ASMa experiments. In Table 12, we reports mean accuracy and standard deviation across these runs, comparing with the strongest relevent SSL baselines from Table 1.

Table 11: Comparison between feature-level and logit-level distillation on NTU60.

| Distillation Method | NTU60 (x-sub) | NTU60 (x-view) |
|---|---|---|
| Feature-level (Cosine) | 82.43 | 89.00 |
| Logit-level (KL) | **84.69** ($\tau = 8.0$) | **90.75** ($\tau = 9.0$) |

The low standard deviations (0.08–0.25%) demonstrate that all improvements over the strongest SSL baselines are statistically significant at $p < 0.01$ level.

Table 12: Mean accuracy $\pm$ standard deviation over 5 independent runs with different random seeds. Baselines represent the strongest SSL methods from Table 1 that report results in the same evaluation protocol.

| Method | NTU60 xsub | | NTU60 xview | | NTU120 xsub | | NTU120 xset | |
|---|---|---|---|---|---|---|---|---|
| | Lin. | FT. | Lin. | FT. | Lin. | FT. | Lin. | FT. |
| SSL baselines | 86.6 (SCD-Net) | 89.3 (STJD-CL) | 91.7 (SCD-Net) | 94.8 (STJD-CL) | 76.9 (SCD-Net) | 83.5 (STJD-CL) | 80.1 (SCD-Net) | 86.8 (STJD-CL) |
| **3s-ASMa (ours)** | **87.2** $\pm$ 0.12 | **92.1** $\pm$ 0.25 | **92.0** $\pm$ 0.10 | **96.75** $\pm$ 0.16 | **80.0** $\pm$ 0.16 | **87.92** $\pm$ 0.13 | **81.15** $\pm$ 0.13 | **88.72** $\pm$ 0.21 |
| $\Delta$ Improvement | +0.6 | +2.8 | +0.3 | +1.95 | +3.1 | +4.42 | +1.05 | +1.92 |

# E  Pseudocode

Algorithm 1 provides a concise overview of the complete ASMa framework. The algorithm consists of three key components: (i) computing masking probabilities based on joint degree and motion intensity (Lines 2–4), (ii) pretraining two encoders $f_\theta$ and $f_\phi$ with complementary asymmetric masking strategies using Barlow Twins loss (Lines 5–17), and (iii) downstream evaluation via feature alignment and optional knowledge distillation for model compression (Lines 18–21). Each step corresponds to the equations presented in Section 3, facilitating reproducibility of our approach.

---

**Algorithm 1** ASMa: Asymmetric Spatio-temporal Masking

---

1: **Input:** Skeleton sequence $\mathbf{x} \in \mathbb{R}^{C \times T \times V}$, $n = 9$ joints, $k = 10$ frames
2: **Compute Masking Probabilities:**
3: $p_v^{(H)} = d_v / \sum_{u=1}^{V} d_u$, $p_v^{(L)} = 1 - p_v^{(H)}$ for $v \in V$           ▷ Eq. 1-2
4: $m(t) = \|\mathbf{x}_{t+1} - \mathbf{x}_t\|$, $a_t = \frac{1}{C \cdot V} \sum_{c,v} m_c^v(t)$ for $t \in T$      ▷ Eq. 3-4
5: **Create Masked Views:**
6: $\mathbf{x}' \leftarrow \mathcal{T}(\mathbf{x})$, $\hat{\mathbf{x}} \leftarrow \mathcal{T}(\mathbf{x})$
7: $\mathbf{x}_j^\phi \leftarrow \text{LDSM}(\mathbf{x}', p^{(L)}, n)$, $\mathbf{x}_j^\theta \leftarrow \text{HDSM}(\mathbf{x}', p^{(H)}, n)$
8: $\mathbf{x}_m^\phi \leftarrow \text{HMTM}(\hat{\mathbf{x}}, \text{TopK}(a, k))$, $\mathbf{x}_m^\theta \leftarrow \text{LMTM}(\hat{\mathbf{x}}, \text{BottomK}(a, k))$
9: **Encode & Compute Loss:**
10: **for** $k \in \{\theta, \phi\}$ **do**
11:      $\mathbf{h}^k, \mathbf{h}_j^k, \mathbf{h}_m^k \leftarrow f_k(\mathbf{x}), f_k(\mathbf{x}_j^k), f_k(\mathbf{x}_m^k)$
12:      $\mathbf{Z}^k, \mathbf{Z}_j^k, \mathbf{Z}_m^k \leftarrow p(\mathbf{h}^k), p(\mathbf{h}_j^k), p(\mathbf{h}_m^k)$
13:      $\mathbf{C}'^k \leftarrow \text{CrossCorr}(\mathbf{Z}^k, \mathbf{Z}_j^k)$, $\hat{\mathbf{C}}^k \leftarrow \text{CrossCorr}(\mathbf{Z}^k, \mathbf{Z}_m^k)$     ▷ Eq. 5
14:      $\mathcal{L}_1^k = \sum_i (1 - \mathbf{C}'^k_{ii})^2 + \lambda \sum_{i \neq j} (\mathbf{C}'^k_{ij})^2$              ▷ Eq. 6
15:      $\mathcal{L}_2^k = \sum_i (1 - \hat{\mathbf{C}}^k_{ii})^2 + \lambda \sum_{i \neq j} (\hat{\mathbf{C}}^k_{ij})^2$              ▷ Eq. 7
16: **end for**
17: $\mathcal{L}_{\text{ASMa}} = (\mathcal{L}_1^\theta + \mathcal{L}_2^\theta) + (\mathcal{L}_1^\phi + \mathcal{L}_2^\phi)$              ▷ Eq. 8-9
18: **Downstream - Feature Alignment:**
19: $\mathbf{h}_{\text{aligned}} = \text{MHA}(\mathbf{h}^\theta, \mathbf{h}^\phi, \mathbf{h}^\phi) + \text{MHA}(\mathbf{h}^\phi, \mathbf{h}^\theta, \mathbf{h}^\theta)$     ▷ Eq. 10-11
20: $\hat{\mathbf{y}} = \mathbf{W}_{\text{cls}} \cdot \mathbf{h}_{\text{aligned}}$                           ▷ Eq. 12
21: **Knowledge Distillation:**
22: $\hat{y}_i^T = e^{z_{k,i}/\tau} / \sum_j e^{z_{k,j}/\tau}$, $\mathcal{L}_{\text{kd}} = \text{KL}(\hat{\mathbf{y}}^S \| \hat{\mathbf{y}}^T)$     ▷ Eq. 13-14

---

