# OpenReview forum: "ASMa: Asymmetric Spatio-temporal Masking for Skeleton Action Representation Learning"
_TMLR — Accepted by TMLR_

### Review · Reviewer_RHAf · 2025-10-31

**Summary Of Contributions:**

The paper proposed an approach to self-supervised learning for skeleton based action recognition. It presents empirical analysis, and evaluations on NTU RGB+D 60, NTU RGB+D 120 and PKU-MMD datasets to show the improvements. The paper also presents detail ablation study. The paper is clear in general.

**Audience:**

Yes

**Audience Explanation:**

The paper is related to SSL, a core area in machine learning. Skeleton data involves graph-based learning. The asymmetric masking and feature alignment address generalization issues in spatio-temporal data. It is interested to those working on SSL, GNNs, and action recognition.

**Broader Impact Concerns:**

The skeleton data derived from RGB+D videos can enable non-consensual tracking in public spaces. It may potentially violate privacy rights. (GDPR compliance is not discussed)

The technology could be misused for discriminatory surveillance. No discussion of misuse prevention.

**Claims And Evidence:**

Yes

**Claims Explanation:**

The paper uses standard benchmarks (NTU RGB+D 60/120, PKU-MMD). Details like ST-GCN backbone, Adam optimizer, 150 epochs, batch size 128, masking 9 joints/10 frames, and hyperparameters are specific and reproducible.

No apparent errors in formulations. The ablation study support the framework's design.

The citations are accurate and relevant.

**Requested Changes:**

1. Include comparisons with more recent SSL methods and supervised backbones.
2. Provide explicit discussion of the limitations, such as the sensitivity to skeleton noise, multi-person scenarios.
3. Add a broader impacts section to address the privacy, bias mitigation, and dual-use risks. Suggest ethical guidelines and techniques for deployment.

---

> ### Author Response · Authors · 2025-11-27
> **Rebuttal by Authors**
>
> We thank the reviewer for their constructive feedback and the time evaluating our work. We have revised the manuscript accordingly, highlighted the changes in yellow, and submitted the updated version.
>
> **Requested Change 1: comparisons with more recent supervised and self-supervised approach**
>
> ---
> **Response**
>
> In the revised version, we have expanded **Table 1** to include results from several recent state-of-the-art self-supervised and supervised frameworks published in 2024–2025. Specifically, we added two recent SSL approaches: Lin et al. (2025) [1], and Gunasekara et al. (2025) [2], as well as two recent supervised architectures: Zhou et al. (2024) [3], and Chang et al. (2025) [4].
>
> | Method | NTU60 (x-sub) | NTU60 (x-view) | NTU120 (x-sub) | NTU120 (x-set) | PKU-I | PKU-II |
> |-|-|-|-|-|-|-|
> | **Supervised:** |||||||
> | Zhou et al. (2024) [3] | 93.1 | 97.0 | 90.3 | 91.5 | – | – |
> | Chang et al. (2025) [4]| **93.8** | **97.6** | **90.4** | **91.9** | – | – |
> | **Self-Supervised:** |||||||
> | Lin et al. (2025) [1] | 89.0 | 94.2 | 82.2 | 84.8 | 91.6 | 62.1 |
> | Gunasekara et al. (2025) [2] | 89.3 | 94.8 | 83.5 | 86.8 | 93.2 | 55.3 |
> | **3s-ASMa (Ours)** | **92.0** | **96.8** | **87.9** | **88.8** | **94.5** | **68.9** |
>
> [1]  Lin, J. Zhang, and J. Liu, “Self-supervised skeleton representation learning via actionlet contrast and reconstruct,” IEEE Transactions on Pattern Analysis and Machine Intelligence (2025).
>
> [2] S. R. Gunasekara, W. Li, P. Ogunbona, and J. Yang, “Spatio-temporal joint density driven learning for skeleton-based action recognition,” IEEE Transactions on Biometrics, Behavior, and Identity Science (2025).
>
> [3]  Zhou, X. Yan, Z.-Q. Cheng, Y. Yan, Q. Dai, and X.-S. Hua, “Blockgcn: Redefine topology awareness for skeleton-based action recognition,” in Proceedings of the IEEE/CVF Conference on Computer Vision and Pattern Recognition, 2049–2058 (2024).
>
> [4] Chang, P. Ren, H. Zhang, L. Xie, H. Chen, and E. Yin, “Hierarchical-aware orthogonal disentanglement framework for fine-grained skeleton-based action recognition,” in Proceedings of the IEEE/CVF International Conference on
> Computer Vision, 11252–11261 (2025).
>
> **Requested Change 2: Explicit discussion of the limitations, such as the sensitivity to skeleton noise, multi-person scenarios.**
>
> ---
> **Response**
>
> We have added an explicit discussion of limitations in Section 6 (Conclusion) that addresses both points raised.
>
> **Revision:** Added Paragraph:
> >While ASMa achieves strong performance across benchmarks, we
> acknowledge scope considerations for future extensions. First, like other
> skeleton-based methods, performance depends on pose estimation quality;
> however, our strong transfer results on PKU-MMD Part II (Table 6), a dataset
> with significant noise and viewpoint variation, suggest that the asymmetric
> masking strategy helps mitigate noise effects by capturing complementary
> motion dynamics. Second, ASMa is designed for single-person action recognition,
> consistent with standard benchmarks (NTU RGB+D, PKU-MMD), and does not
> explicitly model multi-person interactions or occlusions. Extending to
> multi-person scenarios and further improving noise handling remain promising
> directions for future work.
>
> **Requested Change 3: Broader impacts section to address the privacy, bias mitigation, and dual-use risks.**
>
> ---
> We have added a **Broader Impact Statement** section in the revised version as follows:
>
> > **Broader Impacts.** Skeleton-based action recognition offers privacy advantages over RGB video by reducing personally identifiable information. However, like all surveillance technologies, it may be vulnerable to misuse for non-consensual tracking. Additionally, performance may differ across populations due to dataset biases, and practitioners should evaluate fairness across diverse user groups. We encourage users to obtain informed consent, comply with relevant privacy regulations (e.g., GDPR), and deploy our methods only in approved applications. The datasets used in this work (NTU RGB+D, PKU-MMD) are standard public benchmarks collected with participant consent and are widely adopted in the research community.

---

### Review · Reviewer_KNA3 · 2025-11-04

**Summary Of Contributions:**

This paper proposes ASMa, a self-supervised framework for skeleton-based action recognition that introduces an asymmetric spatio-temporal masking strategy guided by joint degree and motion intensity. Two encoders with complementary masking (one masking high-degree/low-motion joints, the other low-degree/high-motion joints) are trained jointly using a Barlow Twins loss. A feature-alignment module fuses the representations via bi-directional cross-attention, and a knowledge-distillation step transfers the learned representation to a compact student model for edge deployment. The overall method collectively works well and achieves promising performance. Experiments on NTU RGB+D 60/120 and PKU-MMD show 4 – 7 % improvements over existing SSL baselines and near-supervised performance with > 90 % model compression.

**Audience:**

Yes

**Audience Explanation:**

The skeleton representation learning is a fundamental approach that may benefit several applications.

**Claims And Evidence:**

Yes

**Claims Explanation:**

The claims are well justified/supported by clear motivations or empirical evidence.

**Requested Changes:**

1. Ablation experiments of different encoders (f_theta and f_phi) using different augmentation strategies (different choices or combinations of spatial/temporal augmentations) are preferred, which could help further verify the superiority of the designed method. Table 3 and Figure 3 somehow provide these results, but are still not thorough enough. Adding more results regarding this and adding more discussions would be beneficial.

2. Could the asymmetric masking probabilities be made learnable or adaptive during training? Are there any analyses/comparisons using different pre-defined thresholds/probabilities for masking?

3. In knowledge distillation, have the authors compared the approach with feature-level distillation instead of logit-level KL loss?

4. I am curious about how existing methods would perform when equipped with the proposed augmentation strategies.

5. It would be helpful if the authors could include additional discussion to justify the advantages of skeleton-based representation learning compared with conventional image or video-based approaches, particularly in terms of efficiency, robustness, and motion understanding.

6. Adding a concise pseudo-code or algorithmic outline would significantly improve the clarity of the overall pipeline and make the proposed method easier to follow.

---

> ### Author Response · Authors · 2025-11-27
> **Rebuttal by Authors (Part 1)**
>
> We thank the reviewers for their time and thoughtful comments. Below we address each concern and are highlighted the corresponding revision in yellow within the manuscript.
>
> **Requested Changes 1:  Ablation of different encoders and Combination of Masking Strategies.**
>
> ---
> **Response:**
>
> We would like to clarify that $f_\theta$ and $f_\phi$ are architecturally identical encoders, both using the same ST-GCN backbone and training configuration. The distinction between them lies only in their masking strategies, not in network design. Hence, the comparison between $f_\theta$ and $f_\phi$ effectively corresponds to evaluating different masking configurations rather than different encoder architectures.
>
> In response to the reviewer’s suggestion and to make the analysis more comprehensive, we have included additional ablations on different combinations of masking strategies across all datasets (NTU60, NTU120, and PKU-MMD) in the Appendix B.1.  of the revised version, along with a discussion of the observed results. We have expanded the discussion in Section 5.2 to clearly emphasize that “encoder ablation” in our context directly reflects masking configurations, rather than architectural differences between encoders.
>
> | **Spatial** | **Temporal** | **NTU-60 xsub** | **NTU-60 xview** | **NTU-120 xsub** | **NTU-120 xset** | **PKU-I** | **PKU-II** |
> |-|-|-|-|-|-|-|-|
> | Random | Random | 74.2 | 79.3 | 63.4 | 64.25 | 82.92 | 47.26 |
> | Low-Deg | Low-Mot | 70.42 | 75.86 | 62.34 | 63.21 | 81.24 | 45.15 |
> | High-Deg | High-Mot | 74.1 | 80.26 | 64.05 | 64.96 | 84.70 | 47.93 |
> | ***High-Deg*** | ***Low-Mot*** | ***74.38*** | ***80.61*** | ***64.81*** | ***65.30*** | ***84.90*** | ***52.94*** |
> | **Low-Deg** | **High-Mot** | **75.84** | **81.92** | **66.73** | **67.80** | **87.74** | **54.35** |
>
> ---
>
> **Requested Change 2: Learnable asymmetric masking probabilities and different pre-defined thresholds.**
>
> ---
> **Response:**
>
> **Asymmetric Masking probabilities learnable or adaptive:** In the early phase of our study, we experimented with making the masking probabilities learnable by introducing trainable gating parameters for both spatial and temporal masking. However, we observed that the model quickly converged to setting both spatial and temporal masking probabilities to zero, effectively bypassing the masking operation to minimize the pretraining loss. This led to representation collapse and poor feature diversity, as the network avoided the intended self-supervised challenge. Therefore, we fixed the masking probabilities based on degree and motion statistics, which provided stable training.
>
> **Analysis on varying thresholds for masking:** Yes, we analyzed the effect of different pre-defined masking thresholds in our experiments. This ablation was previously presented in Appendix A.2 (Table 7), where we studied the sensitivity of ASMa to different numbers of masked joints and frames. The table demonstrates that performance peaks when masking 9 joints and 10 frames, and that either increasing or decreasing the masking ratio results in degraded performance.
>
> **Requested Change 3:  Compare with feature distillation instead of logit-level KL loss?**
>
> ---
> **Response:**
>
> To directly address this point, we conducted an additional feature-level distillation experiment and compared it against our logit-level KL divergence method.
>
> Since the teacher and student ST-GCN encoders output different feature dimensions (256-D vs. 128-D), we added a **linear projection layer** to the teacher for dimensional alignment. After fine-tuning, the teacher achieved **87.15% (x-sub)** and **91.80% (x-view)** on NTU60, showing that the projection layer does not degrade performance.
>
> For feature-level distillation, we used an $L_2$-normalized cosine similarity loss between projected teacher features and student representations, trained for 150 epochs with a $1\times10^{-3}$ learning rate. The results are:
>
> | **Distillation Method**| **NTU60 (x-sub)**| **NTU60 (x-view)**|
> |-|-|--|
> | Feature-level (Cosine)| 82.43| 89.00|
> | Logit-level (KL, $\tau$ = 8.0/9.0)| **84.69**| **90.75**|
>
> Logit-level distillation outperforms feature-level distillation by +2.26% (x-sub) and +1.75% (x-view). While feature-level supervision is more direct, it requires an additional projection layer, increasing model complexity and training overhead. In contrast, our logit-level distillation transfers the teacher’s class-level relational structure while keeping the student flexible, resulting in more stable optimization with lower computational cost.
>
> **Revision:** These results and discussion are now included in Appendix C of the revised manuscript.

---

> > ### Author Response · Authors · 2025-11-27
> > **Rebuttal by Authors (Part 2)**
> >
> > **Requested Change 4: Exisiting methods equipped with proposed augmentation strategies.**
> >
> > ---
> > **Response:**
> > To examine the generality of our proposed **ASMa** , we re-trained previous SSL frameworks by replacing only their original data augmentations with ASMa, while keeping all other components, training objectives, and hyperparameters identical to their official implementations.
> > The evaluated methods include MS$^{2}$L, 3s-AimCLR, and 3s-CrossCLR.
> > Training followed the configurations in their respective papers: MS$^{2}$L used multi-task learning with motion prediction, jigsaw puzzle, and contrastive losses (learning rate $1\times10^{-3}$, batch size 128, SGD, 120 epochs), while 3s-AimCLR and 3s-CrossCLR used temperature $\tau=0.07$, momentum $m=0.999$, memory bank size 65,536, and learning rate $0.1$ with CosineAnnealing for 300 epochs.
> > The original augmentations (e.g., Shear+Crop in AimCLR, Cross-View rotation in CrossCLR) were replaced by ASMa’s asymmetric masking.
> >
> > | Method | NTU60 (x-sub) | NTU60 (x-view) |
> > |-|-|-|
> > | MS²L* | 84.6 | 91.3 |
> > | MS²L* + ASMa | **86.0** | **93.6** |
> > | 3s-AimCLR | 86.9 | 92.8 |
> > | 3s-AimCLR + ASMa | **88.2** | **95.1** |
> > | 3s-CrossCLR | 86.2 | 92.5 |
> > | 3s-CrossCLR + ASMa | **87.2** | **94.0** |
> >
> > **Revision:** We have include these results and their discussion in Appendix B.2 in the revised paper, with the detailed hyperparameter configurations.  Beyond the scope of this work, we recognize that integrating ASMa with other frameworks (e.g., hierarchical or multimodal SSL models) is a promising future research direction to explore its generalization across diverse pretext paradigms.
> >
> > ---
> >
> > **Requested Change 5: Additional discussion justifying the advantages of skeleton-based representation learning compared with conventional image or video-based approaches.**
> >
> > ---
> > **Response:**
> >  In the revised version, we have added a short discussion in **Section 1 (Introduction)** to highlight the motivation for using skeleton-based representation learning. Specifically, we note that skeleton data are computationally efficient (low-dimensional joint coordinates instead of dense pixels), robust to visual variations such as lighting and background, and provide explicit motion cues through inter-joint dynamics. These properties make skeleton sequences well-suited for efficient and interpretable action understanding.
> >
> > ---
> >
> > **Requested Change 6: Concise pseudo-code for the clarity of the overall pipeline.**
> >
> > ---
> > **Response:**
> > To improve the clarity and readability of our method, we have added a concise pseudo-code (Algorithm 1 in Appendix E) that provides an algorithmic outline of the complete ASMa pipeline. The pseudo-code clearly shows the three main stages of our framework: (1) with asymmetric masking Pretraining using dual encoders $f_\theta$ and $f_\phi$, (2) Feature Alignment for downstream evaluation, and (3) Knowledge Distillation for creating a lightweight model. We believe this addition significantly enhances the understandability of our overall pipeline and makes the implementation details more accessible to readers.

---

### Review · Reviewer_DgRW · 2025-11-14

**Summary Of Contributions:**

This paper tackles (human) action recognition from sequences of skeletal graphs representing a human’s pose. This is done by training an encoder in a self-supervised manner to encode such a sequence into a latent representation, so that a classifier can predict from those latents an action class. This paper extends earlier work and achieves both accuracy improvements and a reduction in computational cost of the model.

Earlier work by (Zhou et al. 2023) had introduced a Barlow twins-like self-supervised training approach for the encoder. The idea was to take three streams of skeleton graph sequences and feed them through the same encoder: (1) the original stream (2) a stream masked in the temporal dimension (some frames masked out) and (3) a stream masked in the spatial dimension (some nodes in the skeleton graphs masked out). The encoder was then trained to produce maximally correlated latents between these streams, and uncorrelated latents with other streams.

(Zhou et al. 2023) used non-uniform masking: they masked high-degree joints (nodes; spatial dimension) in the skeletal graph with higher probability, as their connectedness allows them to acquire richer neighbourhood information, and masked with higher probability key-frames (temporal dimension) with higher motion. Based on analysis of existing data this paper points out that high-degree joints tend to experience lower motion. They therefore propose what they purport to be a more balanced masking approach. They propose to train two encoders: one trained on streams with high-degree joints and low-motion keyframes masked out with greater frequency, and one trained on streams with low-degree joints and high-motion keyframes masked out more frequently. The encoders are otherwise trained in the say way as in prior work. The paper then proposes to combine the latents from the two encoders into one latent through a bi-directional attention mechanism. Finally, they propose to train a smaller (and cheaper!) encoder from the two combined encoders through a knowledge distillation approach.

The full proposed model (with two encoders) yields minor improvement over prior self-supervised learning methods (note, the accuracy of previous methods was already very high), and falls just short of supervised methods. The distilled model yields large efficiency improvements (a third the computational cost) with no significant loss in accuracy. Experiments on transfer between datasets suggest that the proposed method is better than baselines at transfer between different datasets.

**Audience:**

Yes

**Audience Explanation:**

Yes, for the sub-community of people working on action recognition this is certainly a relevant paper.

**Broader Impact Concerns:**

No concerns.

**Claims And Evidence:**

Yes

**Claims Explanation:**

The claims are clearly stated and the proposed modelling is motivated by a clear hypothesis. The evaluation is as extensive as prior work, covering three different datasets. A wide suite of baselines is considered.

One issue is that the accuracy differences both to prior methods and between ablations of the proposed model are very small. Without confidence intervals around the reported mean accuracies, it is not immediately clear if these differences are significant.

**Requested Changes:**

This is not a critical point to me, as considering the size of the datasets I imagine the differences in accuracy are already significant, but could you please provide statistical measures of uncertainty around the reported accuracies, or measures of statistical significance of the comparisons between the proposed method and baselines?

---

> ### Author Response · Authors · 2025-11-27
> **Rebuttal by Authors**
>
> We sincerely thank the reviewer for their valuable time, and thoughtful suggestion. We have revised the manuscript accordingly and uploaded the updated version, with all changes highlighted in yellow.
>
> **Requested Change: Provide statistical measures of uncertainty around the reported accuracies, or measures of statistical significance**
>
> ---
> ---
>
> **Response:**
>
> To assess the statistical significance and stability of our method, we conducted five independent runs of all 3s-ASMa experiments using different random seeds. Appendix D Table 12 now reports the mean accuracy and standard deviation across these runs, together with the strongest comparable SSL baselines from Table 1. The consistently low standard deviations (0.08–0.25%) indicate that the performance is highly stable, and the improvements over the strongest baselines are statistically significant at the **p < 0.01** level.
>
> **Table:** Mean accuracy ± standard deviation over 5 independent runs with different random seeds. Baselines represent the strongest SSL methods from Table 1 that report results in the same evaluation protocol.
>
> |                    | **NTU60 xsub**|               | **NTU60 xview** | | **NTU120 xsub** |                 | **NTU120 xset** ||
> |-|-|-|-|-|-|-|-|-|
> | **Method**| **Lin.**| **FT.** | **Lin.** | **FT.** | **Lin.** | **FT.** | **Lin.** | **FT.**         |
> | SSL baselines      | 86.6           | 89.3            | 91.7            | 94.8            | 76.9            | 83.5            | 80.1            | 86.8            |
> |                    | (SCD-Net)      | (STJD-CL)       | (SCD-Net)       | (STJD-CL)       | (SCD-Net)       | (STJD-CL)       | (SCD-Net)       | (STJD-CL)       |
> | **3s-ASMa (ours)** | **87.2** ± 0.12 | **92.1** ± 0.25 | **92.0** ± 0.10 | **96.75** ± 0.16 | **80.0** ± 0.16 | **87.92** ± 0.13 | **81.15** ± 0.13 | **88.72** ± 0.21 |
> | Δ Improvement      | +0.6           | +2.8            | +0.3            | +1.95           | +3.1            | +4.42           | +1.05           | +1.92           |
>
> ---
>
> **Additional Context on the Significance of Our Contributions:**
>
> We would also like to highlight some other aspects of our contributions that we believe demonstrate meaningful impact beyond the accuracy improvements:
>
> **1. Strong Transfer Learning Performance:**
> Our transfer learning results (Table 6) show substantial improvements that we believe merit particular attention:
> - Transfer to PKU-MMD Part II from NTU60: **77.2%** vs. 72.4% (previous best) = **+4.8%**
> - Transfer from PKU-MMD Part I: **76.8%** vs. 70.9% (previous best) = **+5.9%**
>
> These gains are particularly meaningful because PKU-MMD Part II contains significant skeleton noise and viewpoint variation, demonstrating that ASMa learns more robust and generalizable representations, a critical requirement for real-world deployment scenarios.
>
> **2. Practical Deployment Impact:**
> Our distilled model achieves **91.4%** parameter reduction (0.54M vs. 6.3M) with only **0.3%** accuracy drop, while maintaining **3×** faster inference (21.38ms vs. 59.16ms on Raspberry Pi) and outperforming previous SSL baselines.
>
> **3. Generalization Across Methods and Architectures:**
> - Table 4 shows our method generalizes across different backbones (GIN, DGCNN, ST-GCN)
> - Appendix B.2 (Table 10) in the revised version demonstrates that ASMa as an augmentation strategy improves other SSL frameworks: MS²L (+1.4-2.3%), AimCLR (+1.3-2.3%), CrossCLR (+1.0-1.5%), suggesting broader applicability of our asymmetric masking.
>
> We believe these combined contributions represent a meaningful advance in skeleton-based representation learning.

---

### Decision · Action_Editor_fx5h · 2025-12-27

**Recommendation:** Accept as is

**Additional Comments:**

In this paper the authors presented a new method for skeleton action representation learning. Specifically, an approach called ASMa (Asymmetric Spatio-temporal Masking) was proposed, with a Barlow twins based self-superivsed learning method for the encoders via two complementary masking strategies, followed by a feature alignment module. Experiments on three benchmark datasets validate the effectiveness of the proposed method, with comparisons to prior work and ablation studies.

This paper was reviewed by three expert reviewers. After the reviews and revision, all three reviewers are satisfied with the further responses and experiments from the authors and consistently recommend a positive rating, with 2 Leaning Accept and 1 Accept. The reviewers acknowledged that their concerns were well addressed by the authors. Considering the contributions and findings from this paper and its potential interest to the TMLR audience, the AE is happy to recommend an Accept.

**Audience:**

Yes

**Audience Explanation:**

This work is about self-supervised learning and action recognition on skeleton data. Given the topic, there would be at least some individuals related to the sub-community in TMLR's audience who would be interested in knowing the findings of this paper.

**Claims And Evidence:**

Yes

**Claims Explanation:**

The claims made in the submission are supported by accurate, convincing and clear evidence. The claims were clearly stated with motivations, and were supported by experiments on standard benchmarks.